# All-Fiber Laser Feedback Interferometry for Sequential Sensing of In-Plane and Out-of-Plane Displacements

**Zhanwu Xie** **, Meng Zhang, Jie Li, Wei Xia and Dongmei Guo ***

School of Computer and Electronic Information, Nanjing Normal University, Nanjing 210023, China
* Correspondence: guodongmei@njnu.edu.cn

**Abstract:** In this paper, an all-fiber laser feedback interferometer (LFI) with a diffraction grating was developed for sequential measurement of in-plane and out-of-plane displacements without changing the optical arrangement. When the light emitted from an erbium-doped fiber ring laser is incident on a reflection grating at the Littrow angle, the diffracted light will return into the laser cavity along the original path, thus generating laser feedback interference. Experimental results reveal that the all-fiber system could achieve a precision of 40 nm in both in-plane and out-of-plane displacements sensing. Compared with the traditional all-fiber LFI, the proposed sensing system transfers the measuring scale from laser wavelength to grating period, and it has the advantages of good anti-interference performance and reliability.

**Keywords:** all-fiber system; laser feedback interferometer; diffraction grating; displacements sensing



## 1. Introduction

Laser feedback interference occurs when a small portion of light emitted from a laser is reflected or backscattered by an external object and the reflected light reenters the laser cavity, leading to the modulation in both the amplitude and frequency of the lasing field [1,2]. Laser feedback interferometry (LFI) has been widely used in the past decades, such as velocity sensing [3,4], acoustic emission measurement [5,6], micro-displacement sensing [7,8] and quantum communication [9,10]. Among different types of LFI, LFI in the erbium-doped fiber ring laser (EDFRL) is most competitive for coherent fiber-optic sensing applications owing to the benefit of remote optical pumping, immunity to electromagnetic interference, interrogation ability, and the capability of being a wavelength division multiplexed along a single fiber [11,12]. Similar to the conventional optical fiber interferometry, the measurement accuracy of an all-fiber LFI strongly depends on the wavelength stability. As a result, the environmental conditions have to be strictly controlled to achieve high measurement accuracy.

In this paper, we introduce a diffraction grating in an all-fiber LFI system, which transfers the measuring scale from laser wavelength to grating pitch. On the basis of the grating Doppler effect, the proposed system can achieve sequential sensing of in-plane and out-of-plane displacements without changing the optical arrangement. Owing to the low thermal expansion coefficient of the grating substrate, the displacement sensor has good stability and small zero drift. The measurement theory and signal processing are presented. Several experiments are performed to verify the performance of the proposed system. Furthermore, some measurement errors are discussed.

## 2. Principle

### 2.1. Theoretical Analysis of an All-Fiber LFI with a Grating

Figure 1 illustrates the schematic diagram of an all-fiber LFI with a diffraction grating. The pump light with wavelength 980 nm is coupled into the ring cavity through a wavelength division multiplexer (WDM) and then amplified by an erbium-doped fiber (EDF).

The amplified light is transmitted to a $2 \times 2$ optical fiber coupler with a coupling ratio of $\kappa$. The optical isolator is used to ensure the beam one-way transmission in the ring cavity. As seen in Figure 1, $P_{in}$ and $P_{out}$ denote the input and output power of EDF. Remaining cavity losses (isolator, WDM, splices, etc.) are lumped into $\varepsilon_1$ and $\varepsilon_2$. The subscripts $p$ and $s$ represent the pump and signal light, respectively. $P_{seed}$ represents the reflection power from the diffraction grating. The laser beam output from the fiber ring laser is projected onto a reflection grating at angle $\theta$ through a collimator lens (CL), the +1st-order diffracted beam returns along the incident path, thus generating the LFI effect. This angle $\theta$ is called the Littrow angle [13]. Another beam of light is reflected by the fiber Bragg grating (FBG). $P_{laser}$ denotes the LFI signal detected by a photodetector (PD).

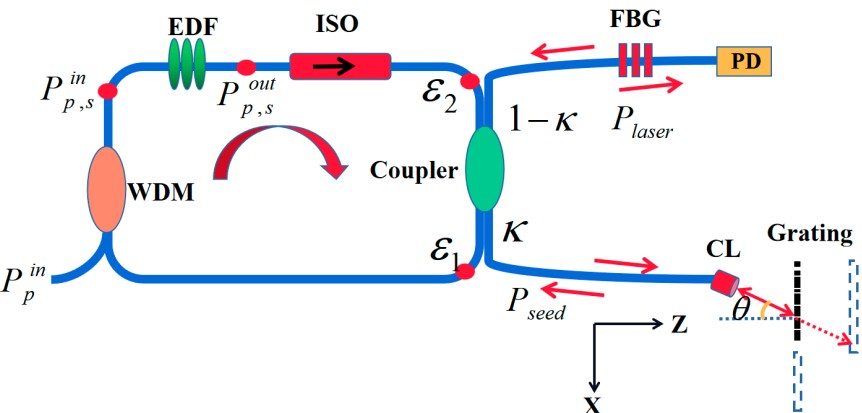

**Figure 1.** Schematic diagram of an all-fiber LFI with a reflection grating. WDM, wavelength division multiplexer; EDF, erbium-doped fiber; ISO, fiber optic isolator; FBG, fiber Bragg grating; PD, photodetector; CL, collimator lens.

According to the diffraction law, incident angle $\theta$ can be expressed as:

$$\theta = \arcsin(\frac{\lambda}{2p}), \tag{1}$$

where $\lambda$ represents the wavelength of the fiber ring laser, and $p$ denotes the pitch of the measured diffraction grating. According to the grating Doppler Effect [14], when the reflection grating moves only in the X-direction with $\Delta x$, a frequency shift is introduced in the diffracted +1st-order beam, and the corresponding phase change of the diffraction beam can be expressed as:

$$\Delta \varphi_g = \Delta \varphi_{gx} = \frac{2\pi \Delta x}{p}. \tag{2}$$

When the diffraction grating moves only in the Z-direction with $\Delta z$ indicated in Figure 1, the +1st-order diffraction beam will generate a phase change due to the grating Doppler effect and a phase change due to the variation of the optical path. The total phase change of the interference signal caused by the out-of-plane displacement $\Delta z$ can be expressed as:

$$\Delta \varphi_g = \Delta \varphi_{gz} = 2\pi \left[ -\frac{\Delta z}{p} \tan\theta + \frac{2\Delta z}{\lambda \cos\theta} \right]. \tag{3}$$

Substituting Equation (1) into Equation (3), the following expression can be obtained:

$$\Delta \varphi_g = \Delta \varphi_{gz} = 2\pi \Delta z \frac{\sqrt{4p^2 - \lambda^2}}{\lambda p}. \tag{4}$$

For the fiber ring cavity, we begin with the amplifier equations for an EDF. These are derived as the steady-state solutions to the rate equations [15,16].

$$P_{P,S}^{out} = P_{P,S}^{in} \exp\left(-\alpha_{P,S}L + \frac{\Delta P_P}{P_{P,S}^s} + \frac{\Delta P_S}{P_{P,S}^s}\right), \tag{5}$$

with

$$\Delta P_S = P_{P,S}^{in} - P_{P,S}^{out}, \tag{6}$$

and saturation powers given by

$$P_{P,S}^s = \frac{hvA_{P,S}^{eff}}{\Gamma_{P,S}\tau(\sigma_{P,S}^e + \sigma_{P,S}^a)}. \tag{7}$$

In Equation (5) through Equation (7), $L$ is the length of EDF. The emission (e) and absorption (a) cross-sections are denoted by $\sigma$, $h$ is the Planck constant, $v$ is the emission frequency which shifts as a periodic function with respect to the external length, $A^{eff}$ is the effective area of the mode, $\alpha$ is the small signal absorption coefficient ($\sigma^\alpha\rho$, where $\rho$ is the Er doping concentration), $\tau$ is the upper state lifetime, and $\Gamma$ is the optical mode–erbium overlap factor.

In the steady state, the power at $P^{out}$ is conserved through one cavity round-trip [17]. We obtain the following expression for $P^{in}$ at the laser wavelength:

$$P_S^{in} = \kappa\varepsilon_1 P_{seed} + (1-\kappa)^2\varepsilon_1\varepsilon_2 r_1 P_S^{out}, \tag{8}$$

where $\varepsilon_1$ and $\varepsilon_2$ denote the total attenuation (e.g., couple, FBG, and splices), and $r_1$ represents the reflectivity of the FBG.

Using Equations (5) and (8), we obtain Equation (9) for one roundtrip.

$$P_S^{out} = (\kappa\varepsilon_1 P_{seed} + (1-\kappa)^2\varepsilon_1\varepsilon_2 r_1 P_S^{out})\exp\left(-\alpha_S L + \frac{\Delta P_P}{P_S^s} + \frac{\Delta P_S}{P_S^s}\right). \tag{9}$$

Taking the logarithm of Equation (5), and solving for $\Delta P_P$, we arrive at the following expression:

$$\Delta P_P = P_S^S\left\{\alpha_S L - \frac{\Delta P_S}{P_S^S} - \ln\left[(1-\kappa)^2\varepsilon_1\varepsilon_2 r_1 + \frac{\kappa\varepsilon_1 P_{seed}}{P_S^{out}}\right]\right\}. \tag{10}$$

Using Equation (5) with (6), we can arrive at another expression for:

$$\Delta P_P = P_P^{in}\left[1 - \exp\left(-\alpha_P L + \frac{\Delta P_P}{P_P^s} + \frac{\Delta P_S}{P_P^s}\right)\right]. \tag{11}$$

Next, $P_S^{out}$ is subtracted from both sides of Equation (8) to arrive at:

$$\Delta P_S = \kappa\varepsilon_1 P_{seed} + [(1-\kappa)^2\varepsilon_1\varepsilon_2 r_1 - 1]P_S^{out}. \tag{12}$$

Finally, Equations (10) and (12) are plugged into Equation (11) to arrive at the transcendental Equation (13).

$$P_S^S\left\{\alpha_S L - \frac{\kappa\varepsilon_1 P_{seed} + [(1-\kappa)^2\varepsilon_1\varepsilon_2 r_1 - 1]P_S^{out}}{P_S^S} - \ln\left[(1-\kappa)^2\varepsilon_1\varepsilon_2 r_1 + \frac{\kappa\varepsilon_1 P_{seed}}{P_S^{out}}\right]\right\}$$
$$= P_P^{in}\left\{1 - \exp\left[-\alpha_P L + \frac{P_S^S}{P_P^s}\left(\alpha_S L - \ln\left((1-\kappa)^2\varepsilon_1\varepsilon_2 r_1 + \frac{\kappa\varepsilon_1 P_{seed}}{P_S^{out}}\right)\right)\right]\right\} \tag{13}$$

where $P_p^{in}$ represents the power of the pump light, $P_s^s$ and $P_p^s$ denote the saturation power of signal light and pump light, respectively, and $P_{seed}$ represents the reflected power from the measured target (reflection grating). The feedback light $P_{seed}$ can be expressed as:

$$P_{seed} = \kappa \varepsilon_2 r^* P_S^{out}. \tag{14}$$

With equivalent reflectance, $r^*$ is:

$$r^* = r_1 + \eta(1 - r_2^2)\cos(\Delta\varphi_g), \tag{15}$$

where $r_2^2$ is the diffraction efficiency of the grating, and $\eta$ is the coupling efficiency from the grating to the CL. Equation (9) is solved numerically for $P_s^{out}$ and the LFI signal detected by PD can finally be indicated as:

$$P_{laser} = \varepsilon_1(1 - \kappa)(1 - r_1^2)P_S^{out}. \tag{16}$$

### 2.2. Numerical Simulation of the LFI Signal Corresponding to Grating Displacement

In order to verify the relationship between the number of interference fringes and the grating displacement, simulations are carried out according to the theoretical analysis. The grating pitch in our measurement system is $p$ = 1.667 μm. Figure 2 presents the simulated results of the LFI signal when the grating moves only in the X-direction. Figure 2a is the simulated sinusoidal displacement of the grating with an amplitude of 8 μm (p–p). Figure 2b is the corresponding LFI signal. Theoretical analysis and simulation results show that each fringe in LFI signal corresponds to the displacement in the X-direction with 1.667 μm. Figure 3 is the simulated results of the LFI signal when the grating moves only in the Z-direction. Figure 3a is the simulated sinusoidal displacement of the grating with an amplitude of 7 μm (p–p). Figure 3b is the corresponding LFI signal. Theoretical analysis and simulation results show that each fringe in LFI signal corresponds to the displacement in the Z-direction with 0.875 μm. The fringe counting method is commonly used for displacement sensing in LFI technology. However, the measurement resolution is only one interference signal. In order to extract the grating displacement with high accuracy, the phase modulation technique is introduced below.

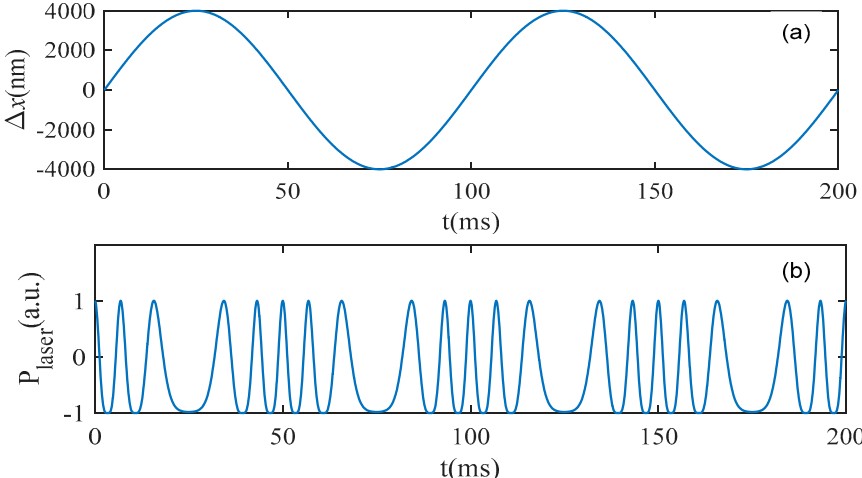

**Figure 2.** Simulation results of the LFI signal corresponding to in-plane displacement $\Delta x$ (8000 nm p–p). (**a**) The simulated displacement of $\Delta x$. (**b**) The simulated LFI signal.

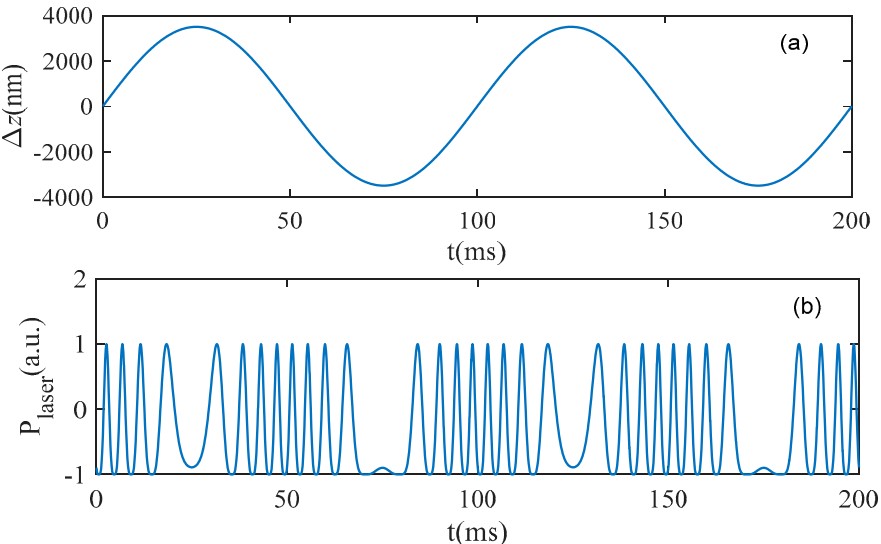

**Figure 3.** Simulation results of the LFI signal corresponding to out-of-plane displacement $\Delta z$ (7000 nm p–p). (**a**) The simulated displacement of $\Delta z$. (**b**) The simulated LFI signal.

### 2.3. Displacement Extraction Method

In order to extract the phase variation corresponding to grating displacement with high accuracy, the sinusoidal phase modulation technique is introduced. Assuming that the phase modulation function is $a\sin(2\pi f_m t)$, where $a$ is the modulation depth, and $f_m$ is the modulation frequency, the alternating component of the modulated interference signal can be obtained using Equation (17).

$$P(t) = P_0 \cos[\Delta\varphi_g + a\sin(2\pi f_m t)], \tag{17}$$

where $P_0$ denotes the output power of the fiber ring laser without feedback. Expanding it in a Fourier series, the harmonics at $f_m$ and $2f_m$ have the following expressions, respectively:

$$P_1(t) = 2J_1(a)P_0 \sin\Delta\varphi_g \sin(2\pi f_m t) = A_1 \sin(2\pi f_m t), \tag{18}$$

$$P_2(t) = 2J_2(a)P_0 \cos\Delta\varphi_g \cos(4\pi f_m t) = A_2 \cos(4\pi f_m t), \tag{19}$$

where $J_n(a)$ is the $n$-th Bessel function. It can be seen from Equations (18) and (19) that $P_1(t)$ and $P_2(t)$ are amplitude-modulated by the sine and cosine functions of the phase $\Delta\varphi_g$. Here, an optimum extraction of the interference signals occurs when the modulation amplitude $a$ is chosen to satisfy $J_1(a) = J_2(a)$. Then, the phase variation corresponding to the grating displacement can be obtained by:

$$\Delta\varphi_g = \arctan[A_1 / A_2]. \tag{20}$$

In the case of in-plane displacement measurement, the displacement $\Delta x$ can be calculated by:

$$\Delta x = \frac{p\Delta\varphi_g}{2\pi}. \tag{21}$$

In the case of out-of-plane displacement measurement, the displacement $\Delta z$ can be obtained by:

$$\Delta z = \frac{\lambda p \Delta\varphi_g}{2\pi\sqrt{4p^2 - \lambda^2}}. \tag{22}$$

## 3. Experimental Results

The experimental setup of an all-fiber LFI system based on a grating is shown in Figure 4. The light output from the erbium-doped fiber amplifier (EDFA) is transmitted to the 2 × 2 coupler with a coupling ratio of 70:30. The fiber optical isolator is utilized to ensure the beam unidirectional transmission in the ring cavity. Port 3 of the coupler is connected to an FBG with 95% reflectivity at wavelength 1550 nm. The optical beam output from port 4 is projected onto a reflection grating at the +1st-order Littrow angle through a CL. The reflection grating with pitch 1.667 μm is fixed on a two axis nano-positioning stage (P762.2L, PI). Both a polarization controller (PC) and a phase modulator (PM) are put between the coupler and the CL.

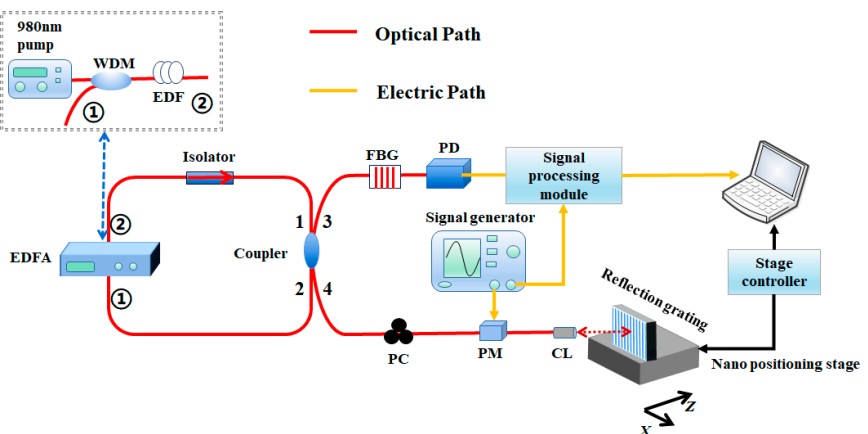

**Figure 4.** Experimental setup of an all-fiber LFI for displacement sensing.

### 3.1. Experimental Observations of the LFI Signal Corresponding to In-Plane or Out-of-Plane Displacement

In order to verify the theoretical and simulation analysis results in Sections 2.1 and 2.2, experimental observations of an LFI signal corresponding to in-plane or out-of-plane displacement were conducted. First, both PC and PM were removed from the system shown in Figure 4. Figure 5 shows the LFI signal corresponding to the in-plane displacement $\Delta x$ of the grating. Figure 5a shows the voltage signal obtained from the X-axis sensor monitor of the stage controller. Figure 5b presents the corresponding LFI signal. This result is consistent with the theoretical simulation results. Figure 6 shows the LFI signal corresponding to out-of-plane displacement $\Delta z$ of the grating. Figure 6a presents the voltage signal obtained from the Z-axis sensor monitor of the stage controller. Figure 6b shows the corresponding LFI signal. It can be seen that the experimental results agree well with the theoretical and simulation analysis.

### 3.2. Sequential Measurement of In-Plane and Out-of-Plane Displacements

In order to evaluate the performance of the all-fiber LFI measurement system, reconstructions of in-plane or out-of-plane grating displacements were conducted. First, the two axis stage was controlled to move sinusoidally only in the X-direction at a frequency of 10 Hz. Figure 7 shows the reconstructed results of sinusoidal motion with amplitude (peak to peak) at 4000 nm, 5000 nm, 8000 nm, and 10,000 nm. In order to evaluate the repeatability of the measurement system proposed, repeated displacement measurements were performed. Table 1 reveals the repeatability of in-plane displacement measurement. Eight measurements were taken for each displacement, and their statistical distributions were analyzed. The mean and standard deviation (SD) of each set of measurements are also given in Table 1.

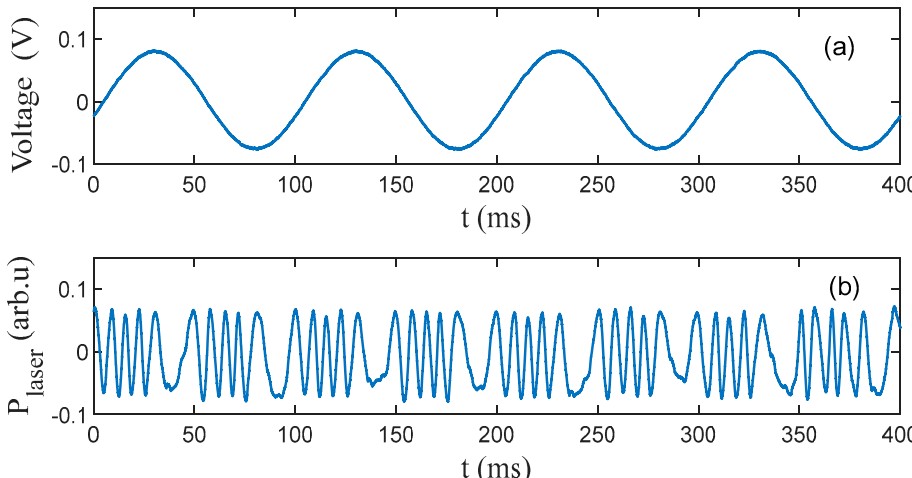

**Figure 5.** Experimental results of the LFI signal corresponding to in-plane displacement $\Delta x$ (8000 nm p–p). (**a**) The voltage obtained from the sensor monitor corresponding to displacement $\Delta x$. (**b**) Experimental LFI signal.

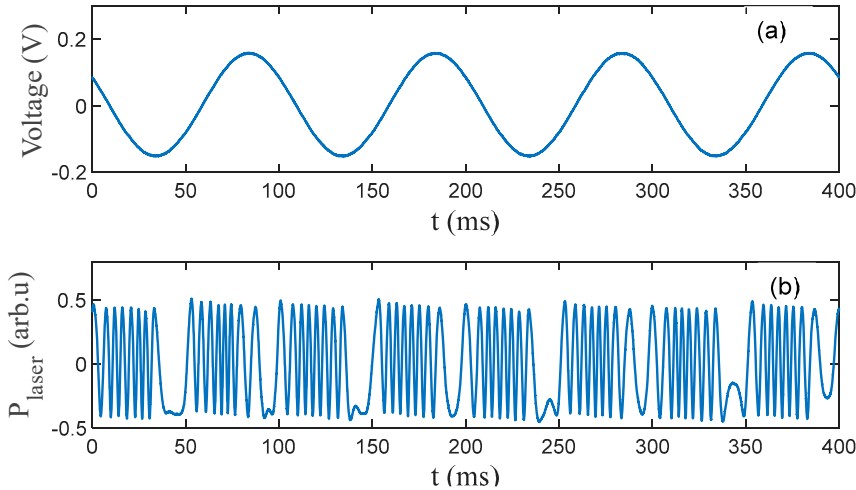

**Figure 6.** Experimental results of the LFI signal corresponding to out-of-plane displacement $\Delta z$ (7000 nm p–p). (**a**) The voltage obtained from the sensor monitor corresponding to displacement $\Delta z$. (**b**) Experimental LFI signal.

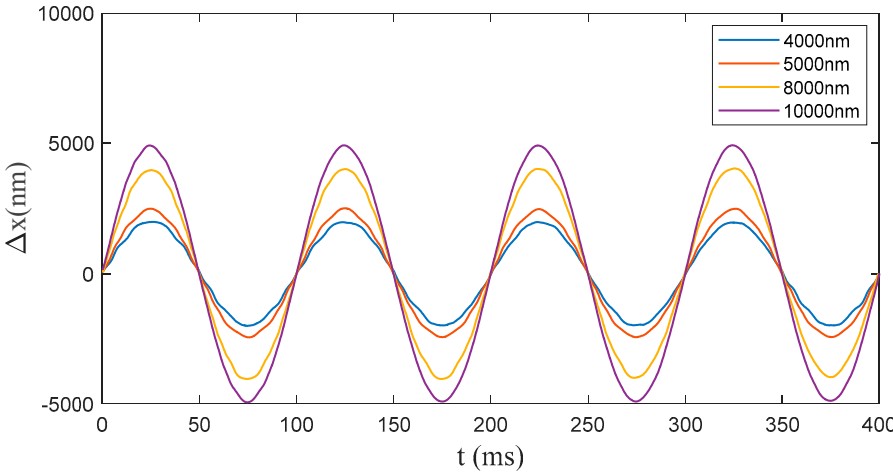

**Figure 7.** Reconstructed results of the sinusoidal motion of grating only in *X*-direction with amplitude (p–p) at 4000 nm, 5000 nm, 8000 nm, and 10,000 nm.

**Table 1.** Measurement results of in-plane displacement $\Delta x$.

| $\Delta x_{\text{p-p}}$ (nm) | 4000 | 5000 | 8000 | 10,000 |
|---|---|---|---|---|
| 1st | 4047 | 4968 | 7994 | 10,031 |
| 2nd | 4051 | 5008 | 7993 | 10,001 |
| 3rd | 4042 | 4964 | 7982 | 9992 |
| 4th | 4031 | 4980 | 7970 | 9981 |
| 5th | 4041 | 4948 | 7916 | 9948 |
| 6th | 4003 | 5032 | 7947 | 9998 |
| 7th | 4019 | 5047 | 7939 | 9995 |
| 8th | 4026 | 4986 | 7949 | 10,029 |
| Mean | 4032.5 | 4994.63 | 7961.25 | 9996.87 |
| SD | 16.09 | 34.55 | 27.98 | 26.41 |

Second, the two-axis nano-positioning stage was controlled to move only in the $Z$-direction with amplitude (peak to peak) at 4000 nm, 5000 nm, 8000 nm, and 10,000 nm. The reconstructed results are shown in Figure 8. Similarly, Table 2 presents the repeatability of the measurements in the Z-direction. From above measurement results, it can be seen that the in-plane and out-of-plane displacements could be measured sequentially with rather satisfying precision and repeatability using the proposed all-fiber system with a grating.

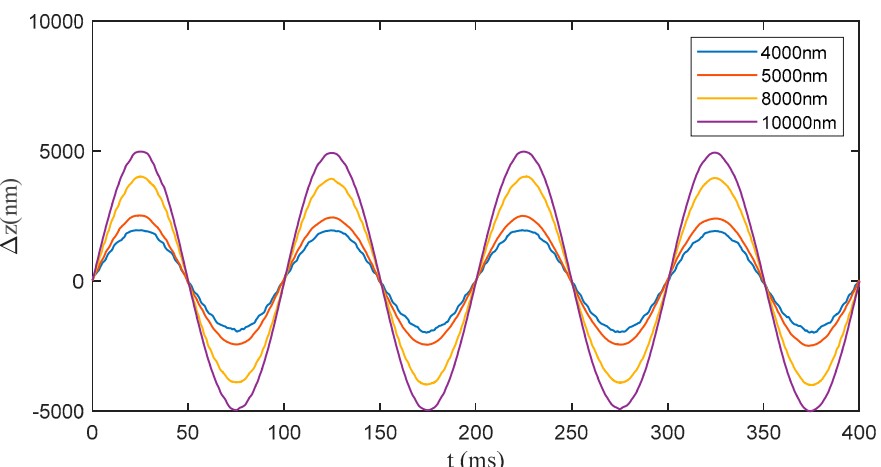

**Figure 8.** Reconstructed results of the sinusoidal motion of grating only in the $Z$-direction with amplitude (p–p) at 4000 nm, 5000 nm, 8000 nm, and 10,000 nm.

**Table 2.** Measurement results of out-of-plane displacement $\Delta z$.

| $\Delta z_{\text{p-p}}$ (nm) | 4000 | 5000 | 8000 | 10,000 |
|---|---|---|---|---|
| 1st | 3949 | 4983 | 7934 | 9973 |
| 2nd | 3956 | 4913 | 7926 | 9918 |
| 3rd | 3939 | 4964 | 7947 | 9949 |
| 4th | 3953 | 4909 | 7981 | 9969 |
| 5th | 3910 | 4978 | 7923 | 9944 |
| 6th | 3960 | 4911 | 8006 | 9850 |
| 7th | 3919 | 4960 | 7922 | 9946 |
| 8th | 3962 | 4910 | 8016 | 9928 |
| Mean | 3943.5 | 4941 | 7956.88 | 9934.97 |
| SD | 19.39 | 33.15 | 38.59 | 38.62 |

## 4. Discussion

### 4.1. Measurement Sensitivity

On the basis of Equation (2) and Equation (4), the measurement sensitivity of in-plane and out-of-plane displacements can be written as:

$$S_x = \frac{d\varphi_{gx}}{dx} = \frac{2\pi}{p}, \tag{23}$$

$$S_z = \frac{d\varphi_{gz}}{dz} = 2\pi \frac{\sqrt{4p^2 - \lambda^2}}{\lambda p}. \tag{24}$$

It can be seen that the measurement sensitivity of in-plane displacement $S_x$ is only dependent on the grating pitch $p$, while the sensitivity of out-of-plane displacement $S_z$ depends on both the grating pitch $p$ and the laser wavelength $\lambda$. In our experiments, the grating pitch was 1.667 μm and the laser wavelength was 1550 nm. Thus, $S_x = 0.216°/\text{nm}$ and $S_z = 0.411°/\text{nm}$ could be obtained from Equations (23) and (24).

### 4.2. Measurement Error

From Equation (21), it can be seen that the theoretical measurement error of the in-plane displacement $|\Delta x|$ is determined by the error of grating pitch $|\Delta p|$ and error of extracted phase $|\Delta\varphi_{gx}|$, which can be expressed as:

$$|\Delta x| = \frac{p}{2\pi}|\Delta\varphi_{gx}| + \frac{\varphi_{gx}}{2\pi}|\Delta p|. \tag{25}$$

Similarly, the theoretical error of the out-of-plane displacement $|\Delta z|$ is influenced by the error of grating pitch $|\Delta p|$, the error of extracted phase $|\Delta\varphi_{gz}|$, and the wavelength variation $|\Delta\lambda|$, which can be deduced from Equation (22).

$$|\Delta z| = \frac{\lambda p}{2\pi\sqrt{4p^2 - \lambda^2}}|\Delta\varphi_{gz}| + \frac{\lambda^3\varphi_{gz}}{2\pi(4p^2 - \lambda^2)\sqrt{4p^2 - \lambda^2}}|\Delta p| + \frac{4p^3\varphi_{gz}}{2\pi(4p^2 - \lambda^2)\sqrt{4p^2 - \lambda^2}}|\Delta\lambda|. \tag{26}$$

Table 3 provides the theoretical measurement error of our system at displacement x = 1000 nm or z = 1000 nm. It can be seen that, when $|\Delta p|$ and $|\Delta\varphi_{gx}|$ (or $|\Delta\varphi_{gz}|$) are better than 1 nm and 1°, and the wavelength variation is better than 1 nm, the theoretical error of in-plane displacement $|\Delta x|$ can be better than 5.23 nm, and the theoretical error of out-of-plane displacement $|\Delta z|$ can be better than 5.84 nm.

**Table 3.** Theoretical measurement error.

|  | Error of Parameters | $|\Delta x|$ | $|\Delta z|$ |
|---|---|---|---|
| $|\Delta\varphi_{gx}|$ (or $|\Delta\varphi_{gz}|$) | 1° | 4.63 nm | 4.86 nm |
| $|\Delta p|$ | 1 nm | 0.60 nm | 0.16 nm |
| $|\Delta\lambda|$ | 1 nm | — | 0.82 nm |
| Total error | — | 5.23 nm | 5.84 nm |

### 4.3. Measurement Speed

As shown above, the phase was extracted from the first harmonic and the second harmonic of the modulated LFI signal. The occurrence of measurement error due to band overlapping in the frequency domain should be considered. The first harmonic and the second harmonic centered at $f_m$ and $2f_m$ presented a spectral width proportional to the maximum velocity of the external target. To avoid overlapping problems, $\varphi_g(t)$ can only have spectral components limited to $f_m/2$, which can be expressed as:

$$\frac{d\varphi_g(t)}{2\pi dt} \leq \frac{f_m}{2}. \tag{27}$$

Substituting Equations (2) and (4) into Equation (27), the following relationships can be obtained:

$$v_x \leq \frac{p f_m}{2},$$ (28)

$$v_z \leq \frac{\lambda p}{2\sqrt{4p^2 - \lambda^2}} f_m,$$ (29)

where $v_x$ and $v_z$ are the velocity of the in-plane and out-of-plane displacement, respectively. It can be seen that the maximum measurable velocity of the in-plane displacement depends on the grating pitch and the modulation frequency, while the maximum measurable velocity of the out-of-plane displacement relies on the laser wavelength, grating pitch, and the modulation frequency.

**5. Conclusions**

In summary, we proposed an all-fiber LFI based on a reflection grating for sequential measurement of in-plane and out-of-plane displacements without changing the optical arrangement. The in-plane displacement is traceable to the grating pitch, while the out-of-plane displacement depends on the grating pitch and the laser wavelength. A series of experiments were performed to demonstrate the performance of our proposed system. Theoretical analysis showed that the measurement precision of the system could achieve $p/318$ in in-plane sensing and $p/285$ in out-of-plane sensing. The measurement precision can be improved by further refining the system's electronics and mechanics. This method has advantages such as high sensitivity and a compact structure. It can be a useful sensor to monitor the displacement and vibration of the precision motorized stage in a wide variety of research applications. Considering more industrial applications, our further research will focus on the simultaneous measurement of in-plane and out-of-plane displacements using an all-fiber LFI based system.

**Author Contributions:** Conceptualization, D.G., Z.X. and W.X.; methodology, M.Z.; software, J.L. and M.Z.; validation, D.G.; writing—original draft preparation, Z.X.; writing—review and editing, D.G., Z.X. and W.X.; visualization, D.G.; supervision, D.G. and W.X.; project administration, D.G.; funding acquisition, D.G. All authors have read and agreed to the published version of the manuscript.

**Funding:** This work was supported by the National Natural Science Foundation of China (51875292).

**Institutional Review Board Statement:** Not applicable.

**Informed Consent Statement:** Not applicable.

**Data Availability Statement:** Not applicable.

**Conflicts of Interest:** The authors declare no conflict of interest.

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
