# Peer review of "All-Fiber Laser Feedback Interferometry for Sequential Sensing of In-Plane and Out-of-Plane Displacements"

_photonics, doi:10.3390/photonics10040480_

Round 1

Reviewer 1 Report

The authors introduced an all-fiber feedback interferometer for sequential measurement of in-plane and out-of-plane dispalcement. The overall quality is good and acceptable for publication on Photonics.

Author Response

    Thank you for your recommendation. We have double checked the whole manuscript to avoid grammatical errors , typos and other mistakes.

Reviewer 2 Report

The submission titled “All-fiber laser feedback interferometry for sequential sensing of in-plane and out-of-plane displacements” developed an all-fiber laser feedback interferometer (LFI) with a diffraction grating, which can be used for sequential measurement of in-plane and out-of-plane displacements without changing the optical arrangement. The authors analyzed several factors that might affect measurement performance. Experimental results show that a resolution of 40 nm can be achieved in both in-plane and out-of-plane displacements. The work presented in this paper provides a promising method for displacement sensing in narrow and harsh environments. However, the article has some logical problems, and the arguments need further elaboration. Some major revisions are need to further clarify the content of this submission before it can be considered for publication.

1) The authors wrote “However, when the target is in a narrow and harsh environment, there are some difficulties in using LFI sensing system in free space.” in Introduction, which is the advantage of all-fiber configuration not the proposed scheme. However, before this sentence, the author has already begun to refer to his own protocol and there should be the advantages of the proposed scheme. The introduction needs to be revised to show the advantages and disadvantages of all-fiber system and the benefits of your own system

2) LFI, one kind of interferometer, has been widely used, and the citations of relevance are far from sufficient. Interferometers can also be applied in quantum communication, and some references [r1-r5] are suggested to be cited.

[r1] Laser Photon. Rev. 6, 393-417 (2012);

[r2] Sci. Bull. 67, 2167-2175 (2022);

[r3] Opt. Lett., 46, 821-824 (2021);

[r4] J. Lightw. Technol. 39, 4217-4224 (2021);

[r5] PRX Quantum 3, 020315 (2022);

3) Can you give a further explanation about Fig. 2? Most of the description of the amplitude seems to be in the later part of the article. The structure of the text needs to be modified.

4) In practical situations, it is almost impossible for all displacements to be in-plane and out-of-plane. Can the authors give a further discussion about the condition that the direction of movement is inclined at an angle with the xz plane, for example, at 45 degrees?

5) The whole paper needs to be double proofread, as there are some mistakes in the manuscript, for example, SD’s full name is missed. In addition, the resolution of Fig. 4 is not very high.

Author Response

Thank you very much for your comments and suggestions. We have carefully addressed your insightful comments and suggestions, responded to these comments and suggestions point-by-point, and revised the manuscript accordingly. Please see the attachment. 

Author Response

Thank you for your recommendation.  We have carefully responded your comments and suggestions point-by-point, and revised the manuscript accordingly. Please see the attachment. 

Round 2

Reviewer 2 Report

The authors addressed my comments and I recommend it for publication.